

# Integration of the metabolome and transcriptome reveals indigo biosynthesis in *Phaius flavus* flowers under freezing treatment

Yi-Ming Zhang[1,2,3], Yong Su[2], Zhong-wu Dai[1,3], Meng Lu[1,2,3], Wei Sun[2], Wei Yang[2], Sha-Sha Wu[1,3], Zhi-Ting Wan[1,3], Hui-Hua Wan[2] and Junwen Zhai[1,3]

[1] College of Landscape Architecture, Fujian Agriculture and Forestry University, Fuzhou, China
[2] Institute of Chinese Materia Medica, China Academy of Chinese Medical Sciences, Beijing, China
[3] Key Laboratory of National Forestry and Grassland Administration for Orchid Conservation and Utilization, Fuzhou, China

## ABSTRACT

**Background**. Indigo-containing plant tissues change blue after a freezing treatment, which is accompanied by changes in indigo and its related compounds. *Phaius flavus* is one of the few monocot plants containing indigo. The change to blue after freezing was described to explore the biosynthesis of indigo in *P. flavus*.

**Methods**. In this study, we surveyed the dynamic change of *P. flavus* flower metabolomics and transcriptomics.

**Results**. The non-targeted metabolomics and targeted metabolomics results revealed a total of 98 different metabolites, the contents of indole, indican, indigo, and indirubin were significantly different after the change to blue from the freezing treatment. A transcriptome analysis screened ten different genes related to indigo upstream biosynthesis, including three anthranilate synthase genes, two phosphoribosyl-anthranilate isomerase genes, one indole-3-glycerolphosphate synthase gene, five tryptophan synthase genes. In addition, we further candidate 37 cytochrome P450 enzyme genes, one uridine diphosphate glucosyltransferase gene, and 24 $\beta$-D-glucosidase genes were screened that may have participated in the downstream biosynthesis of indigo. This study explained the changes of indigo-related compounds at the metabolic level and gene expression level during the process of *P. flavus* under freezing and provided new insights for increasing the production of indigo-related compounds in *P. flavus*. In addition, transcriptome sequencing provides the basis for functional verification of the indigo biosynthesis key genes in *P. flavus*.

# INTRODUCTION

Natural indigo is one of the oldest dyes, and its use has been documented in India as early as 2600 BC (*Gaboriaud-Kolar, Nam & Skaltsounis, 2014*; *Balfour-Paul, 1998*). Indigo is a commodity with great economic significance, and the only source of indigo dye is plants containing indigo (*Balfour-Paul, 1998*; *Clark et al., 1993*; *Sequin-Frey, 1981*). The

Corresponding authors
Hui-Hua Wan, hhwan@icmm.ac.cn
Junwen Zhai, zhai-jw@163.com

extraction of indigo from plants was completely replaced when the chemical pathway to indigo synthesis was discovered in the 1870s by Adolf von Baeyer. Since then, plant indigo dye has been completely replaced by synthetic indigo (*Daykin, 2011*). However, artificial production of indigo requires the synthesis of a potential carcinogen sodium dithionite, which produces environmental problems. Therefore, extracting indigo from plants has attracted attention again (*Garcia-Macias & John, 2004*). Indigo and related compounds also have important value in the medical field. For example, indigo and indirubin are both potent aryl hydrocarbon receptor ligands for treating cancer; indirubin also has a significant inhibitory effect on white blood cells; and isatin has antibacterial, anti-tumor, and neuroprotective properties (*Adachi et al., 2001*; *Pandeya et al., 2005*). A variety of compounds in the indigo metabolic pathway have important economic and medicinal value. Therefore, studying the plant indigo biosynthetic pathway to increase the yield of indigo compounds has become a hot spot in phytochemistry research.

*Phaius flavus* belongs to Collabieae (Orchidaceae) with bright-coloured flowers and has medicinal and ornamental value (*Xiang et al., 2014*). When *Phaius* is physically, chemically or biologically damaged, the wound will turn blue in a short time. Studies have shown that the substance causing this blue-changing phenomenon is indigo (*Lüning, 1967*). Thus, *P. flavus* is considered a potential plant to produce industrial dye (*Xia & Zenk, 1992*). Indigo has a special distribution in nature and only exists in a few families such as: fabids: *Isatis* (Brassicaceae); *Crotalaria* and *Indigofera* (Leguminosae); Malvids: *Polygala* (Polygalaceae) and a base group of superasterids: *Polygonum* (Polygonaceae); Asterids: *Wrightia* and *Echites* (Apocynaceae); *Marsdenia* and *Asclepias* (Asclepiadaceae); *Clerodendrum* (Verbenaceae); *Baphicacanthus* (Acanthaceae) (*Huang, Pang & Zhou, 2010*); as well as *Calanthe* s. l., *Bletia* and *Epidnedium* (Orchidaceae), among other (*Xia & Zenk, 1992*; *Zhou & Duan, 2005*). Among them, Orchidaceae are the only monocotyledonous group reported to contain indigo, so its uniqueness deserves more attention.

The indigo synthesis pathway belongs to the branch of the tryptophan pathway, originated from indole, and is catalyzed by enzymes of the cytochrome P450 monooxygenase (CYP450) to form indoxyl (3-hydroxyindole) (*Warzecha et al., 2007*). Indoxyl is toxic to cells. Indigo plant for protection of its own cells use UDP-glucosyltransferase (UGT) to catalyze the glycosylation of indoxyl to form a colorless and harmless indican and is stored in vacuoles (*Daykin, 2011*). When the plant is damaged, the vacuole compartment disappears, and the indican in the vacuole is reduced to indoxyl by $\beta$-glucosidase (GLU) in the chloroplast, and then oxidized by the oxygen in the air to dimerize to form blue indigo, and indirubin is formed by the condensation of isatin and indoxyl (*Minami et al., 1997*; *Papanastasiou et al., 2012*).

We have previously measured the content of indolyl derivatives (isatin, indican, indigo, and indirubin) in the flowers and leaves of six *Calanthe* alliance plants including *P. flavus* before and after a freezing treatment that changes the plants to blue. The results showed that the key compounds in the indigo pathway changed significantly after a freezing treatment (*Zhang et al., 2020*). Therefore, to further explore the metabolism and molecular mechanisms of the *P. flavus* freezing, this study used non-targeted and targeted metabolomics to explore the changes in the overall and key compounds in *P. flavus* before

and after the freezing treatment. The transcriptomics approach have been used to analyze the global expression of genes related to indigo biosynthesis. Further, we used the real-time quantitative polymerase chain reaction (qPCR) to verify the key gene expression patterns of indigo biosynthesis before and after the freezing treatment. This study provides candidate genes for future verification of indigo-related functional genes in *P. flavus*.

## MATERIALS & METHODS

### Plant material

The experimental materials were taken from the forest orchid garden of Fujian Agriculture and Forestry University (Fuzhou, 260°5′20″N, 1191°3′45″E). The winter is short and the summer is long in this area of China. The annual average temperature is 16–20 °C (range, −6–42.3 °C). The relative humidity is 77% (*Huang, Liu & Ma, 2018*). The flowering period of *P. flavus* was in late April 2019 in the testing garden. The *P. flavus* flowers in full bloom were randomly selected to simulate mechanical damage through quick freezing in ultra-low temperature. Concretely, the flowers were immersed in liquid nitrogen for just 2 s, placed at room temperature for 10 min as the freezing treatment sample. The flowers without treatment were used as the control group (CK). Each group of samples were divided into two parts, one part was used for metabolite detection, and the other part was for RNA isolation. Three biological replicates were set for each treatment (Fig. 1) Three biological replicates were set for each treatment (Fig. 1).

### Extraction and analysis of non-targeted metabolites

Non-targeted metabolomic analysis of CK (M-normal-Q) samples and the blue-changing after freezing (M-injure-H) samples, each treatment set of three biological replicate experiments. The freeze-dried sample powder (100 mg) was extracted with 1.2 mL of 70% methanol overnight at 4 °C and was vortexed six times to improve extraction efficiency. Subsequently, after centrifugation at $10,000 \times g$ for 10 min, the clear liquid was collected with a pipette and filtered with a 0.22 μm microporous membrane, and stored in a sample bottle for ultra-high performance liquid chromatography-tandem mass spectrometry (UPLC-MS/MS; for UPLC: Shim-pack UFLC SHIMADZU CBM30A; Shimadzu, Shanghai, China for MS/MS: Applied Biosystems 6500 QTRAP; Applied Biosystems, Beijing, China analysis. The quantitative detection of metabolites was monitored using triple quadrupole (QQQ) multiple reaction monitoring (MRM) mode.

A ACQUITY UPLC HSS T3 C18 column (2.1×100 mm, 1.8 μm; Waters Corp., Milford, MA, USA) kept at 40 °C was used as stationary phase. The binary mobile phases were water (A) and acetonitrile (B), both of which comprised 0.04% acetic acid. The gradient program was set as follows: 5%–95% B(0–10 min); 95% B (10–11 min); 95%–5% B (11–11.1 min); 5%B (11.1–14 min). The flow rate and injection volume was set as 0.35 mL/min and 2 μL, respectively.

Mass conditions: positive ion mode; ESI ion source temperature 550 °C; mass spectrometry voltage 5,500V; curtain gas 30 psi; and the CAD parameters for collision-induced ionization were set to high. Each particle pair for triple quadrupole QQQ detection was based on declustering pressure and collision energy (*Chen et al., 2013*).

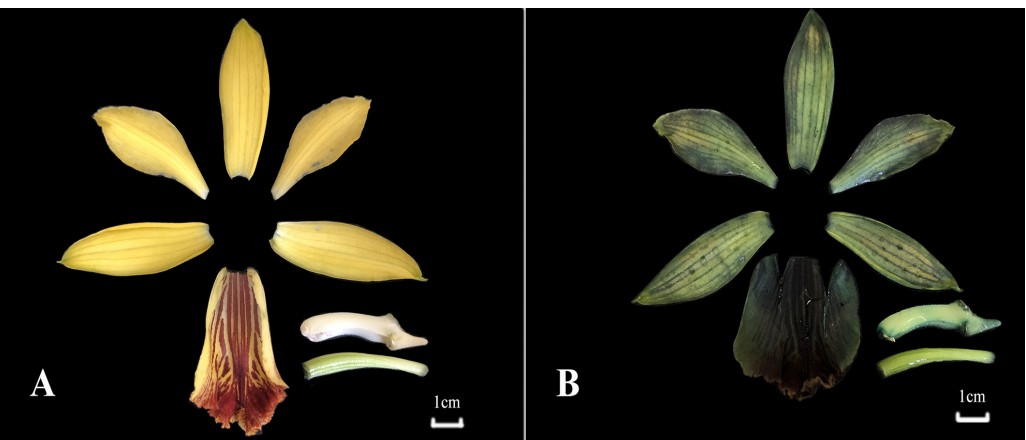

**Figure 1** **The _P. flavus_ flower before freezing treatment and blue-changing after freezing treatment.**
(A) Before freezing treatment; (CK); (B) blue-changing after freezing treatment.

The metabolites were quantified according to the standard comparison of the _m/z_ value, retention time, and fragmentation mode of the secondary spectrum information in the database compiled by MetWare Biotechnology Co., Ltd. (Wuhan, China) (_Yang et al., 2019_). The filter conditions for significantly changed metabolites (SCM) were: |log2 (fold change)|>1 and _p_-value <0.05. R (http://www.r-project.org/) was used to perform a principal component analysis (PCA) on the SCM to study the specific accumulation of metabolites.

### Extraction and detection of indigo compounds by targeted metabolites

The indigo (CAS: 482-89-3) and indirubin (CAS: 479-41-4) standards were procured from Shanghai Yuanye Biotechnology Co., Ltd. (Shanghai, China), with purity >99%; the indican (CAS: 487-60-5) were purchased from TCI Chemical Development Co., Ltd. (Shanghai, China), with purity >98%. Indole was purchased from Sigma-Aldrich (Shanghai, China), with purity >97%.

A 20 mg portion of _P. flavus_ flower powder was accurately weighed and extracted with 10 mL DMF by ultrasonication for 30 min in triplicate. The one mL sample solution was dried in a nitrogen blower and then redissolved in 1ml methanol solution. Subsequently, it was filtered through a 0.45 μm membrane and used for UPLC-MS/MS analysis.

Triple quadrupole mass spectrometry (Agilent 6410; Agilent Technologies, Palo Alto, CA, USA) was performed using the indigo, indirubin, indican, and the indole quantitative reference method. The chromatographic column was an Agilent EclipsePlus C18 column (2.1 × 100 mm, 1.8 μm), the mobile phase A comprised water containing 0.2% formic acid (A) and mobile phase B comprised only acetonitrile (B), with a linear gradient elution at a flow rate of 0.3 mL/min gradient elution. The elution gradient was set as follows: 5% B (0–2 min); 5–27% B (2−−5.5 min); 27–20%B (5.5–7.5 min); 20–25% B (7.5–10 min); 25–55% B (10–20 min); 55–100% B (20–20.1 min); and then 5%B balance to 22 min. Use MRM mode to detect target compounds and the charge ratio (_m/z_) of the product

in the positive ion mode was: indigo 263.0815 →132; indirubin 263.0815 →219; indican 296.1129 →134.

Indole was quantified by GCMS-QQQ, and the chromatographic conditions were referenced from the determination of the volatile components in *Dendrobenthamia japonica* (*Bai & YU, 2019*). one mL of the filtrate was dried under nitrogen gas, and 200 μL of methanol was added for reduction. Before injecting the sample for GCMS-QQQ analysis, it was passed through a 0.45 μm membrane filter. The chromatographic column was a TG-5MS (30 m ×0.25 mm ×0.25 μm); the flow rate of the helium carrier gas was 1.0 mL/min; inlet temperature was 250 °C, the split ratio was 10:1, and injection volume was 1 μL; the initial temperature of the oven was maintained for 5 min, then the temperature was raised to 190 °C at the rate of 3 °C/min; temperature was maintained for 3 min, and then raised to 230 °C at a rate of 10 °C/min, when the temperature was maintained for 5 min; the mass spectrometer was used in full scan mode to collect the signals. The ionization source was EI, ionization energy was 70V, and the scanning range of the mass spectrum was 30–400 *m/z*.

## cDNA library construction and sequencing

Extraction of RNA were carried out from treated and control *P. flavus* flowers using the plant RNA rapid extraction kit (RN38 EASY spin PLUS; Beijing Adler Biotechnology Co., Ltd., Beijing, China; RN38 EASY spin PLUS) as per the manufacturer's instructions. The concentration and purity of the RNA were measured with the Nanodrop 2000 spectrophotometer, and the Agilent 2100 instrument was used to assess RNA integrity. Samples with RNA integrity number (RIN) >7 and concentrations ≥ 100 ng/ μL were taken for cDNA library construction. cDNA library construction was performed using 5ug total RNA and NEB Next Ultra RNA Library Prep Kit from Illumina. The cDNA library was checked for quantity and quality with Qubit 2.0 (Life Technologies, Carlsbad, CA, USA) and Agilent Bioanalyzer 2100 system respectively. Then sequencing was carried out using Illumina HiSeq 2500 platform.

## Transcriptome assembly and annotation

Before assembling and data analysis, raw reads were screened to remove the adapters, the reads of low sequencing quality (the number of bases with a quality value of Q20 accounted for >50% of all reads), reads with N proportion >10%, and the duplicates, to obtain high-quality clean reads. Trinity was used to break the sequence into fragments (K-mer). The K-mer fragment was extended to obtain a longer fragment (Contig), the overlap between the fragments was used to obtain a collection of fragments, and the sequencing information of the reads and the reconstructed Brujin graph method were used to assemble and identify the transcripts (*Bryant et al., 2017*). The Corset hierarchical clustering comparison was used to perform hierarchical clustering of the reads from the transcripts to obtain a de-redundant unigene, thereby obtaining ALL-Unigenes of *P. flavus* flowers.

Blastx (*E*-value <1× $10^{-5}$) was used to compare all assembled unigenes sequences with the protein databases, including the Kyoto Encyclopedia of Genes and Genomes (KEGG,

https://www.genome.jp/kegg/), the National Center for Biotechnology Information non-redundant (NR, ftp://ftp.ncbi.nih.gov/blast/db/), Swiss-Protein (https://www.uniprot.org/), Gene Ontology (GO, http://geneontology.org/), and the clusters of orthologous groups for eukaryotic complete genomes (COG/KOG, ftp://ftp.ncbi.nih.gov/pub/COG/KOG/kyva) databases. Various pathway annotations of the unigenes were obtained through the KEGG annotation information. Based on the NR annotation results, Blast2GO software was used to obtain the GO annotation information for each unigene. We used WEGO software to perform the GO functional classification and statistical visualization of all unigenes, and then used Trinotate to perform an association analysis to obtain the comparative annotation results (*Ye et al., 2006*). Finally, the functional distribution of the *P. flavus* unigenes was obtained.

## Potential *DEGs* expression analysis

To assess the expression levels of each unigene before and after freezing *P. flavus* flowers, clean reads of each sample were mapped to the assembled unigenes using Bowtie2 software. The FPKM (Fragments Per Kilobase of transcript per Million fragments mapped) method was used to normalize the expression of the unigenes to eliminate the effect of the difference in sequencing amount and length of the different unigenes on the calculated expression amount. edgeR (*Robinson, McCarthy & Smyth, 2010*) was used to analyze the differences in expression before and after the freezing treatment. FDR <0.05 and |log2FC|>1 were judged as *DEGs*, which were screened to obtain *DEGs* of the flowers before and after the freezing treatment. We further mapped the *DEGs* to reference pathways in the KEGG database, and combined the changes in the indigo-related metabolites with the expression changes of the related *DEGs* in the pathways to more accurately reflect the candidate genes related to indigo synthesis.

## Validation of the *DEGs* by qRT-PCR

We randomly selected 8 candidate genes related to indigo biosynthesis for qRT-PCR analysis to verify the transcriptome data results. Gene-specific primer pairs for 150–250 bp regions were designed for different genes (Table S1). There are three biological replicates for each gene analyzed by qRT-PCR. Actin was selected as the reference gene (*Jiang et al., 2017*). All qRT-PCR reactions were performed using *TrannsStart* [®] Top Green QPCR SupeMix kit (Beijing Quanjin Biotechnology Co., Ltd., Beijing, China) according to the manufacturer's instructions.

## RESULTS

### Non-targeted metabolomics analysis of changes in the main compounds of *P. flavus*

The CK (M-normal-Q) samples and the freezing blue-changing treated (M-injure-H) samples were clearly divided into two clusters, indicating that the broad-target metabolomics detection data were stable and reliable (Fig. 2A). All 627 compounds were detected in the control and treatment groups, which were divided into 22 categories. Metabolites with fold change $\geq$ 2 and fold change $\leq$ 0.5 were selected as significantly

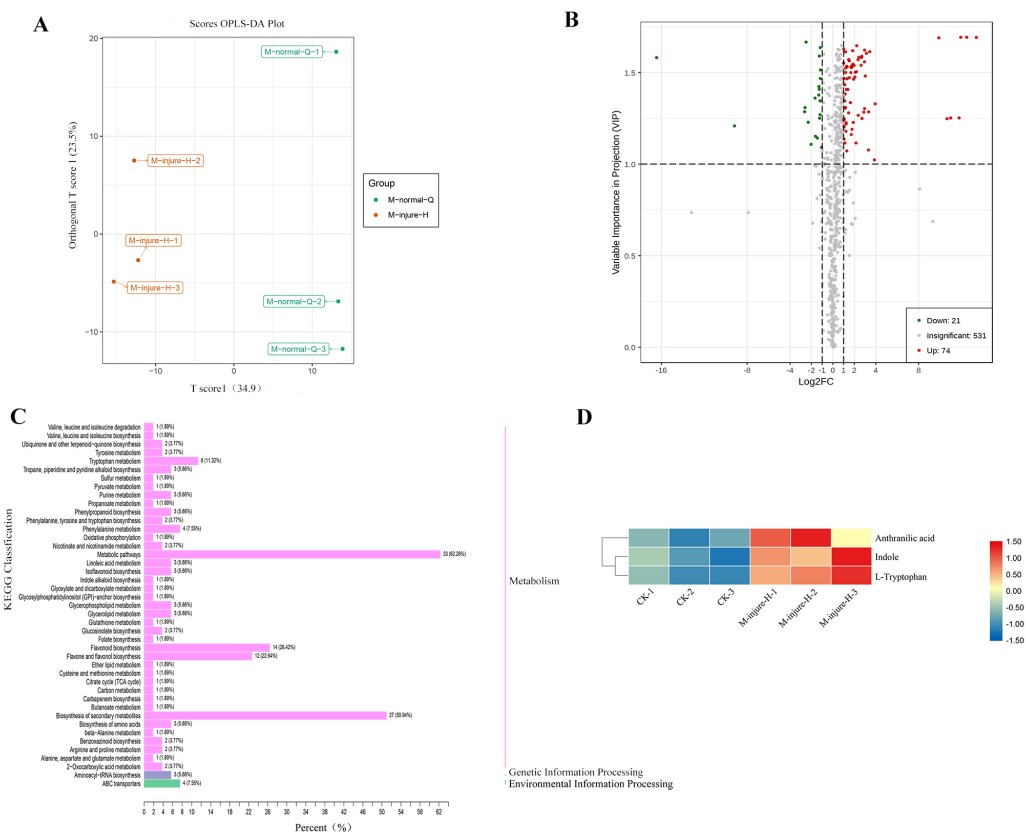

**Figure 2 Analysis of non-targeted metabolomics.** (A) PCA score plot metabolite profiles from CK(M-normal-Q) and blue-changing after freezing treatment (M-injure-H). (B) The volcano map of P. flavus between CK and blue-changing after freezing treatment. red dots show the increased content, green dots show the decreased content, and gray dots show the insignificant change; (C) Classification map of KEGG differential metabolites CK and blue-changing after freezing treatment of *P. flavus*. (D) Indole, Anthranilic acid and L-Tryptophan heat map of *P. flavus* between CK and blue-changing after freezing treatment, the color of the grid represents the level of substance content, red represents higher substance content, and blue represents lower substance content.

different metabolites. Based on the OPLS-DA analysis and the results obtained for the variable importance in project (VIP), metabolites with a VIP value ≥ 1 were selected as significantly different. The results showed that 95 different metabolites were screened from the 627 metabolites. Among them, 74 metabolites were upregulated and 21 metabolites were downregulated. See Fig. 2B for details. The KEGG enrichment analysis of the 95 differential metabolites mapped the metabolites to 43 pathways, which were divided into Metabolism (86.79%), Genetic Information Processing (5.66%), and Environmental Information Processing (7.55%) (Fig. 2C). Further analysis revealed that the 95 metabolites were divided into 13 categories including: Lipids, Organic acids and derivatives, Indole derivatives, Vitamins and derivatives, Terpene, Alkaloids, flavonoid, Nucleotide and derivatives, Phenolamides, Polyphenol, Phenylpropanoids, Amino acid and derivatives, and other of which flavonoids were the main compound followed by lipids.

Peer J

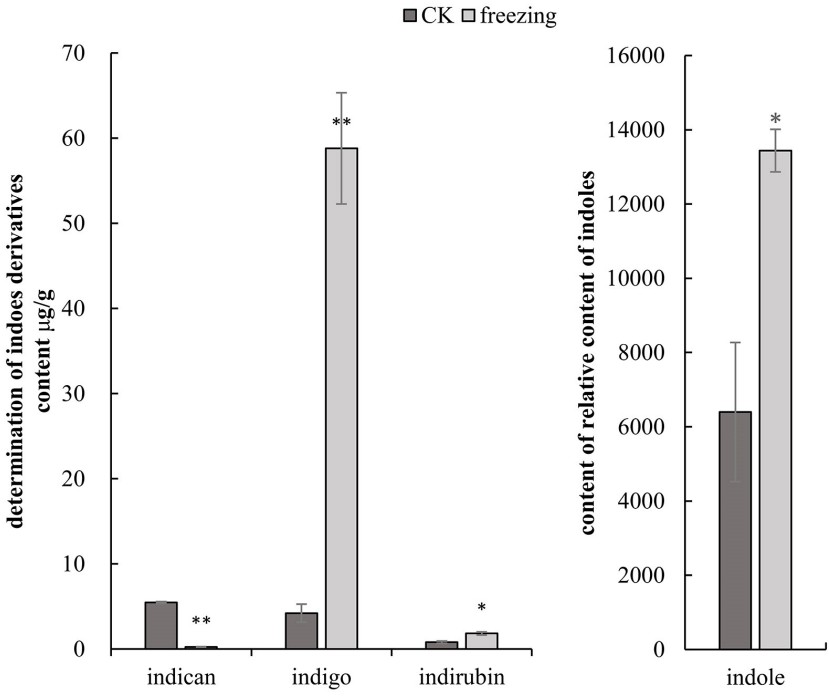

**Figure 3 Targeted metabolomics to determine changes in indigo-related compounds.** (A) Changes in content of indican, indigo and indirubin CK and blue-changing after freezing treatment in *P. flavus* by LCMS-QQQ; (B) The change of indoles content blue-changing after freezing treatment in *P. flavus* by GCMS-QQQ, the error line indicates the standard deviation of three biological repeats, and the significant difference analysis is shown by analysis of $T$-test, *: $P < 0.05$, with significant difference; **: $P < 0.01$, with extremely significant difference.

Three substances closely related to the indigo pathway were found among the 95 different metabolites, and showed an upward trend, including anthranilic acid, L-tryptophan, and indole (Fig. 2D). The presence of these compounds suggests that the freezing blue-changing treatment may affect the change in indigo content.

## Targeted metabolomics analysis of indigo-related compound changes in *P. flavus*

The three-quadrupole LCMS-QQQ method was used to determine the contents of indican, indigo, and indirubin to clarify their changes related to indigo synthesis before and after the *P. flavus* freezing treatment. GCMS-QQQ was used to determine the relative indole content. The results show that indican had a good linear relationship at $Y = 2385.5X + 215.85$ ($R^2 = 0.9996$), indigo at $Y = 1248.7X + 241$ ($R^2 = 0.9999$), and indirubin at $Y = 666.24X + 370.2$ ($R^2 = 0.9992$). The results showed that indole, indigo, and indirubin contents increased significantly by 2.1-, 14-, and 2.3-fold before and after the freezing treatment ($P < 0.05$). The indigo glycoside content decreased extremely significantly by 22.8-fold ($P < 0.01$) (Fig. 3).

**Table 1  Statistical table of annotation unigenes in *P. flavus*.**

| Database | Number of genes | Percentage (%) |
|---|---|---|
| KEGG | 121802 | 39.54 |
| NR | 163637 | 53.12 |
| SwissProt | 90730 | 29.45 |
| Trembl | 161990 | 52.58 |
| KOG | 91015 | 29.54 |
| GO' | 97986 | 31.81 |
| Pfam | 100356 | 32.57 |
| Total unigenes | 308077 | 100 |

## Sequencing assembly and annotation of the transcriptome

In order to screen the gene expression and changes related to indigo biosynthesis before and after freezing of *P. flavus*, the CK group was named normal-Q-1, normal-Q-2, and normal-Q-3 before the freezing treatment and injury-H-1, injury-H-2, and injury-H-3 after the freezing treatment. The cDNA library was prepared and RNA-seq sequencing was performed on the two groups of samples. *De novo* splicing technology combined with Trinity was used to mix and assemble the samples to construct the *P. flavus* reference gene library, which had 58.76 Gb of CleanData. Each sample had 7 Gb of CleanData and 307,739 unigene reference genes. The quality assessment showed that the N50 length was 1,446 bp, GC content was 44.262%, Q20 percentage was 97%, Q30 was 93%, and all unigenes are in the range of 200-3,000 bp in length. All base numbers compared to N50 were 278,947,003 nt. To sum up, the quality of the transcriptome sequencing was high and met the requirements for an in-depth analysis.

All-Unigenes of *P. flavus* obtained by sequencing were assessed with the KEGG, NR, Swiss-Prot, Termbl, Pfam, GO, COG/KOG databases ($E$-value $<1\times 10^{-5}$). A total of 308,077 unigenes corresponding functional annotation information were obtained, of which 163,637 (53.12%) unigenes were annotated in the NR database; 90,730 (29.45%) unigenes were obtained in the Swiss-port database; 121,802 (39.54%) unigenes were annotated in KEGG database; 91,015 (29.54%) unigenes were annotated in KOG database; 161,990 (52.58%) unigenes were annotated in Trembl database; 100,356 (32.57%) unigenes were annotated in the Pfam database; there were 164,749 unigenes (53.48%) that had an annotation in at least one database (Table 1).

The GO functional annotation results showed that 97,686 unigenes were classified into three categories of biological processes, cellular components, and molecular functions. These unigenes included 58 biological functions. Among them, the unigenes involved in biological processes and cellular components were the main components. In the biological processes, the proportion of unigenes that the cellular processes was the highest with 65,264 unigenes annotated, followed by metabolic processes and biological regulation, which annotated 59,455 and 23,744 unigenes respectively. The highest proportion of unigenes participated in cells and cell parts, with 68,608 and 68,507 unigenes, respectively in the cell component category. In the molecular function category, unigenes accounting

for the highest proportions were two types of binding and catalytic activation, with 63,031 and 56,666 unigenes, respectively (Fig. S1).

To better understand the biological functions and pathways represented by each unigene, KEGG annotated 121,802 unigenes mapped to 141 biological signaling pathways, including metabolism, genetic information processing, environmental information processing, and many other biological metabolic pathways (Table S1). Among them, the top three with the most unigenes were metabolic pathways with 20,746, biosynthesis of secondary metabolites with 9,668, and RNA transport with 2,963.

### Differentially expressed gene (*DEG*) enrichment analysis

A total of 17,134 differentially expressed genes were screened based on differential gene expression analysis, of which 10,805 were up-regulated and 6,329 were down-regulated. Based on the results of GO and KEGG enrichment analysis, the *DEGs* before and after the freezing blue-changing treatment of *P. flavus* were selected.

The results of the GO cluster analysis showed (Fig. 4A) that the genes included 54 biological functions, including 28 biological processes, 16 cellular components, and 10 molecular functions. A total of 147,134 unigenes were detected in biological processes, of which metabolic processes and cellular processes accounted for the highest proportions, with 6,161 and 6,768 unigenes, respectively. The proportion of *DEGs* in cell and cell parts of cellular components was the highest, with 7,926 and 7,911 unigenes, respectively. Binding and catalytic activity in molecular functions represented the highest proportion of *DEGs*, with 6,277 and 5,925 unigenes, respectively.

A total of 12,226 *DEGs* were obtained by KEGG enrichment and 141 metabolic pathways were compared. The indigo pathway starts with indole, so indole-related biosynthesis is significantly related to indigo. The indole alkaloid synthetic pathway contained 8 unigenes; the tryptophan metabolism pathway contained 75 unigenes; and the phenylalanine, tyrosine, and tryptophan biosynthetic pathway contained 228 unigenes. The top 20 metabolic pathways in the KEGG enrichment analysis (Fig. 4B).

### Screening of genes related to indigo biosynthesis

BLASTN was used to compare and search the *P. flavus* transcriptome database according to the annotation results of the protein databases to identify the relevant functional genes in *P. flavus* before and after the freezing treatment as comprehensively as possible. A total of 73 unigenes that may be related to indigo biosynthesis were screened through the metabolic map and unigene annotation results. Among them, nine DEGs were candidate for involvement in indole biosynthesis, including three anthranilate synthase (AS) genes (Cluster-38184.203518, Cluster-38184.63640, Cluster-38184.74434), two phosphoribosylanthranilate isomerase (PAI) genes(Cluster-38184.13823, Cluster-38184.137190), one indole-3-glycerol phosphate synthase (IGPS) gene(Cluster-38184.112686), five tryptophan synthase (TS) genes (Cluster-38184.103420, Cluster-38184.136314, Cluster-38184.139029, Cluster-38184.171387, Cluster-38184.55751), which were all significantly upregulated after freezing. Thirty-seven cytochrome enzyme monooxygenase (CYP450) genes, two uridine diphosphate glucosyltransferase (UGT)
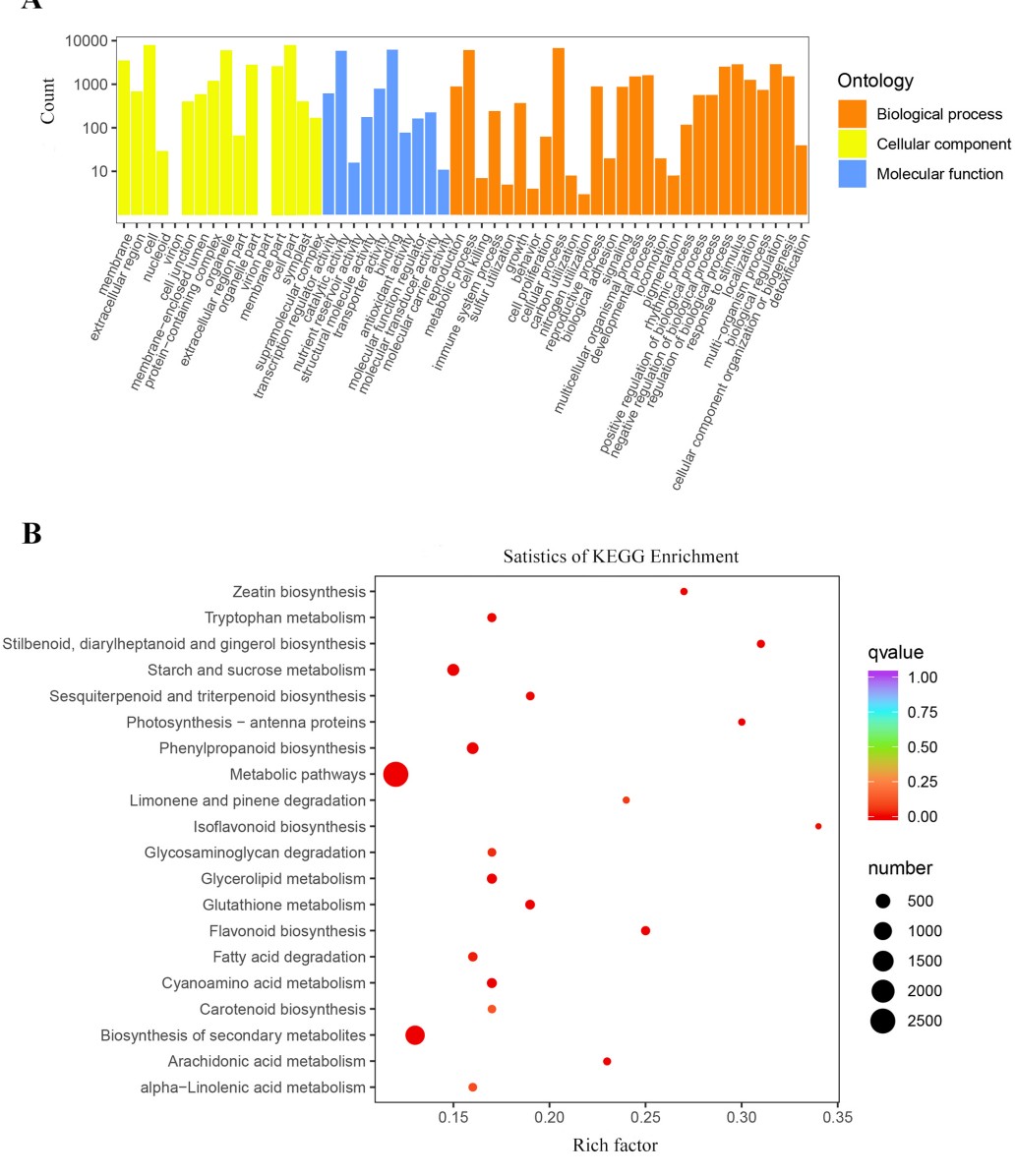

**Figure 4 Enrichment analysis of differential genes.** (A) Classification of the GO for the different expression Unigenes in *P. flavus*; (B) KEGG enrichment of the top 20 pathways among different expression genes $P < 0.05$.

genes, and 24 $\beta$-D-glucosidase (GLU) genes were jointly involved in the production of indigo (Fig. 5 and Table S3).

## qRT-PCR validates *DGEs* related to indigo biosynthesis

We performed qRT-PCR on eight representative unigenes related to indigo biosynthesis to verify the *DEGs* obtained by RNA-Seq Primer design (Table S2). The expression levels of *ASA* (Cluster-38184.203518), *PAI (* Cluster-38184-137190), *IGPS (* Cluster-38184.112686*)*, *TSA* (Cluster-38184.103420), *CYP71A4* (Cluster-38184.135548), *UGT1*

**Figure 5** **The biosynthetic pathway of indigo compounds in *P. flavus* predicted based on the results of transcriptome annotation.** The dotted arrow indicates the synthesis pathway of indigo and indirubin; The number represents the number of candidate genes, and the red and green respectively represent the up-regulation and down-regulation of genes before and after the freezing blue-changing treatment. (ASA, anthranilate synthase alpha; ASB, anthranilate synthase beta; PAI, phosphoribosylanthranilate isomerase; IGP, indole-3-glycerolphosphate synthase; TSA, tryptophan synthase alpha; CYP, cytochrome P450 enzymes; UGT, uridine diphosphate glucosyltransferase; GLU, $\beta$-D-glucosidase genes).

(Cluster-38184.119005), and *GLU* (Cluster-38184.103035) were upregulated after the freezing treatment compared to before the freezing treatment, *UGT2* (Cluster-12095.0) was down-regulated after the freezing treatment compared to before the freezing treatment, which identify with the expression results of the transcriptome (Fig. 6 and Table S3). It shows that the transcriptome data is credible.

# DISCUSSION

## Metabonomic changes of *P. flavus* before and after freezing treatment

In this study, 95 different compounds were screened before and after the *P. flavus* freezing treatment through non-targeted metabolomics. It includes 13 categories: Lipids, Organic acids and derivatives, Indole derivatives, Vitamins and derivatives, Terpene, Alkaloids, Flavonoid, Nucleotide and derivates, Phenolamides, Polyphenol, Phenylpropanoids, Amino acid and derivatives, among others. Among them, the main category is flavonoids. Flavonoids are an important compound in the plant immune system, which play an important role in plant stress resistance (*Saito et al., 2013*; *Nakabayashi et al., 2014*). Therefore, most of the flavonoids changed significantly after freezing treatment. It is worth noting that both flavonoids and indigo compounds belong to the branch of the shikimic pathway. The shikimic pathway is a bridge connecting primary and secondary metabolism. It is widely present in plants. Studies have shown that there is a certain synergistic change between the shikimic acid pathway and flavonoid metabolism (Li,

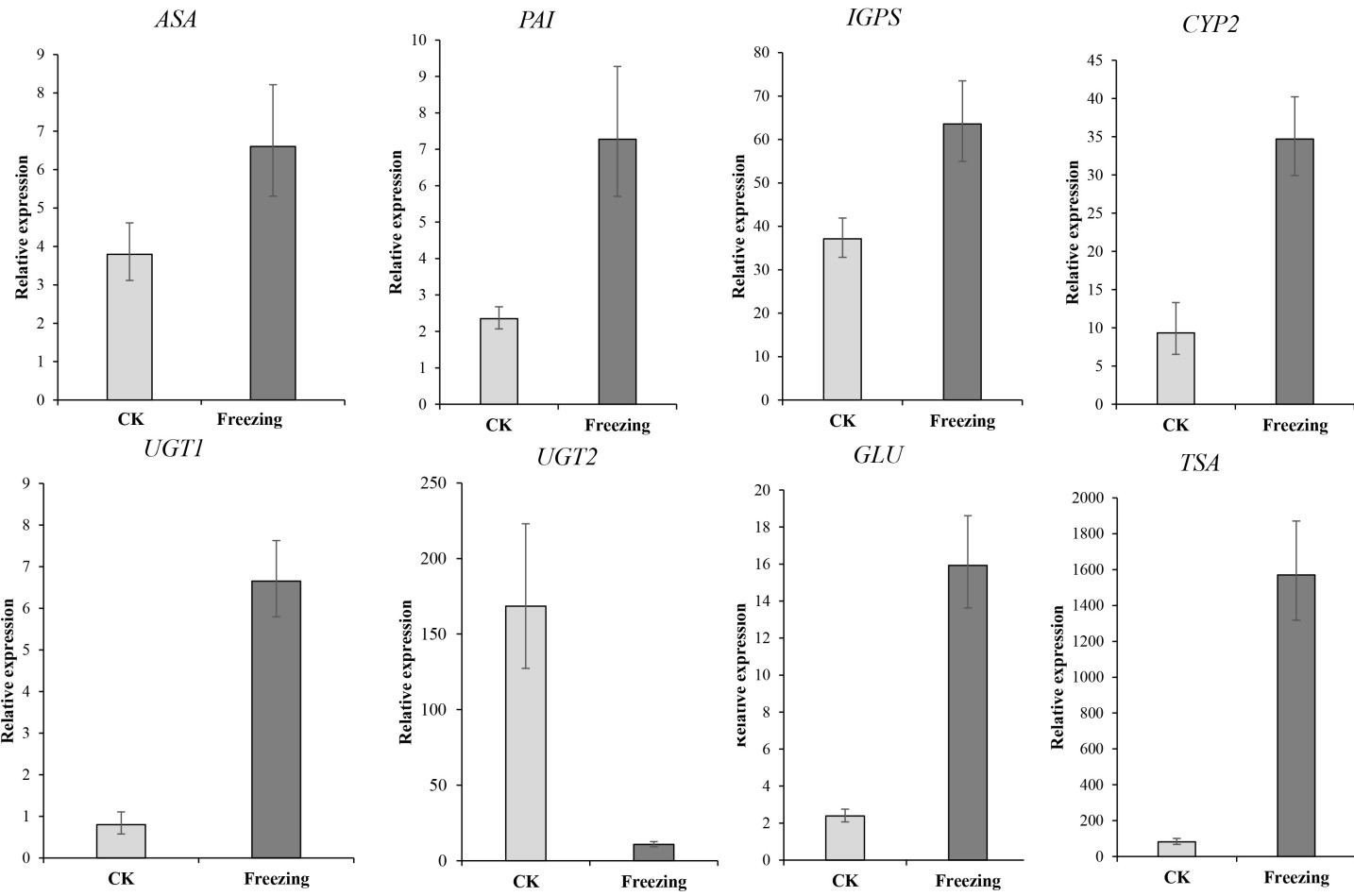

**Figure 6** **The qRT-PCR verification of the expression profile of *P. flavus* flowers CK and blue-changing after freezing treatment.** Use the $2^{-\triangle\triangle Ct}$ to calculate the relative expression level of Cluster-38184. 203518 (ASA), Cluster-38184-137190(PAI). Cluster-38184.112686 (IGPS), Cluster-38184.135548 (CYP2(CYP71A4)), Cluster-12095.0 (UGT1), Cluster-38184.119005 (UGT2), Cluster-38184.103035 (GLU), Cluster-38184.103420 (TSA). Error bar indicates SD.

2016). Similarly, according to the theory of metabolic flow, the metabolic changes of flavonoids are likely to affect the metabolic accumulation of indigo. As the research on the indigo pathway is still unclear, the MetWare database (*Chen et al., 2013*) lacks indigo compound information, but we can still find some clues about the changes in upstream compounds related to indigo biosynthesis.

The indigo biosynthetic pathway starts with indole and follows the shikimic acid metabolic pathway (*Xia & Zenk, 1992*; *Jin et al., 2016*). Among them, the contents of anthranilic acid and indole, indigo precursors, tended to increase. This may be why the contents of indigo and indirubin increased, which is the end product of metabolism. To further explore the mechanism of the change in the indigo-related compounds before and after the freezing treatment, it is necessary to conduct an in-depth analysis and discussion on the DEGs identified by the transcriptome.
## Functional genes related to initiation substrate of indigo biosynthesis

Indole is the initial product of the indigo pathway, which follows the shikimic acid and tryptophan metabolic pathways. Indole biosynthesis directly affects indigo and indirubin contents (*Xia & Zenk, 1992*; *Chen et al., 2018*). In the shikimic metabolic pathway, shikimic acid undergoes a series of complex reactions and is converted to chorismate, which is catalyzed by anthranilate synthase to form anthranilic acid. Anthranilic acid is catalyzed by PAT and PAI to form 1-(2-carboxyphenylamino)-1-deoxy-D-ribulose 5-phosphate, which is catalyzed by indole-3-glycerol phosphate synthase (IGPS) to form indole precursor indole-3-glycerophosphate (IGP) (*Jin et al., 2015*). IGP is catalyzed by the tryptophan synthase (TS) alpha subunit (TSA) to form indole, and the beta subunit (TSB) combines indole with serine to form tryptophan (*Li et al., 2020*).

Studies have reported that upregulation of the AS, PAI, IGPS, and TS genes increases tryptophan and indole contents (*Chen et al., 2018*; *Gao et al., 2016*; *Jin et al., 2016*; *Li et al., 1995*; *Zeng et al., 2016*). Nine *DEGs* related to indole biosynthesis were candidates in the transcriptome database. These genes included three AS genes, two PAI genes, one IGPS gene, and five TSA genes. All of these genes were upregulated in *P. flavus* after freezing. The GCMS-QQQ results showed that indole tended to increase in *P. flavus* flowers after freezing, which was identified with the above gene expression results (Figs. 3 and 5). That is to say, the high expression of genes related to indole biosynthesis, such as, PAI, IGPS, and TS, provides more precursors for indigo biosynthesis, thereby indirectly increasing the content of indigo-related compounds.

## Functional genes related to downstream products of indigo biosynthesis

Hydroxylation of indole is the key step in the indigo pathway. Many studies have indicated that cytochrome P450 monooxygenase participates in the hydroxylation of indole (*Song, 2011*; *Schullehner et al., 2008*). In this study, 37 CYP450 monooxygenase-upregulated genes were annotated in the transcriptome data ($P < 0.01$ Table S3). The LCMS-QQQ and GCMS-QQQ results showed that indole, indigo, and indirubin contents increased significantly ($P < 0.05$, Fig. 3), indicating that the high expression of CYP450 monooxygenase promotes the conversion of indole to indoxyl, thereby producing more indigo in the freezing tissues of *P. flavus* flowers. Recently, *Inoue, Morita & Minami (2021)* have reported that Flavin-monooxygenase can also achieve indole hydroxylation. In the reported hydroxylation of indole, the hydroxylation of indole is consistent from the perspective of substrate and product, but the enzymatic reactions mediated are diverse. In other words, this catalytic process may be completed by different kinds of enzymes in different species. The key enzyme in the indigo biosynthesis pathway is indoxyl-uridine diphosphate glucose (UDP)-glucosyltransferase (UGT) (*Inoue et al., 2017*; *Inoue et al., 2018*; *Marcinek et al., 2000*; *Wang et al., 2019*). The reason is that indoxyl has a toxic effect on plants (*Kim et al., 2009*). Indoxyl is glycosylated to form a stable colorless precursor substance indican, which is stored in vacuoles, and protect cells from indoxyl (*Daykin, 2011*; *Inoue et al., 2017*; *Yuan, Liu & Xiao, 2014*). Recent research has found that *PtUGT*,

which has been identified in *Polygonum tinctorium*, can realize the glycosylation of indoxyl (*Hsu et al., 2018*). In this study, two homologous *PtUGT* transcripts with high homology were selected from the transcript by comparison, one is the downregulated Cluster-38184.119005 and the aonther is upregulated Cluster-12505.0 (Table S3). According to the metabolic flux theory, we cannot determine the gene expression level of genes related to indican is up-regulated or down-regulated during the freezing process. Therefore, the gene expression of indican needs further functional verification. Studies have shown that the conversion of indoxyl to indican in plants is a reversible reaction. When the plant is damaged, the cell compartmentalization disappears, the indican in the vacuoles are released, and it is hydrolyzed by the *β*-glucosidase and reduced back to indoxyl. Finally, indoxyl is oxidized and dimerized by air to form the indigo and indirubin (*Nabors, Stowe & Epstein, 1967*; *Minami et al., 1999*; *Jin et al., 2016*). Twenty-four DEGs in the transcriptome encoding upregulated *β*-D-glucosidase were identified (Table S3). These above-mentioned candidate genes together promoted the enrichment of indoxyl, which increased the indigo and indirubin contents in *P. flavus* tissue after the freezing treatment.

Overall, significant differences in indole, indican, indigo, and indirubin contents were detected in *P. flavus* flowers after the freezing blue-changing treatment. Transcriptome and metabolite analysis indicated that genes related to indole synthesis were upregulated after the freezing treatment, resulting in an increase in indole content. During the process of frostbite, the up-regulated CYP450 continues to catalyze the formation of indoxyl from indole, and together with the indoxyl formed by hydrolysis of indican to form more indigo and indirubin. These results suggest that the transformation of indigo after freezing is not only a biochemical change, but a dynamic change involving genes. Freezing treatment may damage the cell lipid membrane and/or cell wall of *P. flavus* flower, but DNA, RNA, and protein are still alive, and they can continue to express genes or catalyze the formation of indigo.

## CONCLUSIONS

The effect of freezing treatment on indigo biosynthesis in *P. flavus* flowers was revealed based on experimental data. The metabolome consequence showed that the contents of anthranilic acid, indole, indigo, and indirubin increased significantly after the freezing treatment. The transcriptome analysis combined with the metabolomic analysis showed that key genes in the indigo biosynthetic pathway, such as PAI, IGPS, TSA and CYP were upregulated. In addition, qRT-PCR verified the expression levels of these genes. These results indicate that the significant increase in indigo was due to the high expression of related genes after freezing and the combination of hydrolysis and oxidation of related precursor compounds, and not simply a biochemical reaction. In addition, this study screened several genes related to indigo biosynthesis and provided a pool of candidates for future research on the functional verification of key genes in indigo biosynthesis in *P. flavus*.

## ACKNOWLEDGEMENTS

The authors would like to thank TopEdite for its linguistic assistance during the preparation of this manuscript. Thanks to Zexing Li and Bugeng Agricultural Technology Co., LTD for the collection and preservation of experimental materials.

### Funding

This work research was funded by National Key R&D Program of China (2021YFE1011900, 2019YFC1711100) Scientific and Technological Innovation Project of China Academy of Chinese Medical Sciences (CI2021A04112); Opening project of Shanghai Key Laboratory of Plant Functional Genomics and Resources (PEGR202204) and supported by Disciplinary Professional Construction Project of College of Arts and College of Landscape Architecture (YSYL-bdpy-2). The funders had no role in study design, data collection and analysis, decision to publish, or preparation of the manuscript.

### Grant Disclosures

The following grant information was disclosed by the authors:
National Key R&D Program of China: 2021YFE1011900, 2019YFC1711100.
Scientific and Technological Innovation Project of China Academy of Chinese Medical Sciences: CI2021A04112.
Opening project of Shanghai Key Laboratory of Plant Functional Genomics and Resources: PEGR202204.
Disciplinary Professional Construction Project of College of Arts and College of Landscape Architecture: YSYL-bdpy-2.

### Competing Interests

The authors declare there are no competing interests.

### Author Contributions

- Yi-Ming Zhang conceived and designed the experiments, performed the experiments, analyzed the data, authored or reviewed drafts of the paper, and approved the final draft.
- Yong Su performed the experiments, prepared figures and/or tables, authored or reviewed drafts of the paper, and approved the final draft.
- Zhong-wu Dai, Meng Lu, Wei Yang, Sha-Sha Wu and Zhi-Ting Wan analyzed the data, authored or reviewed drafts of the paper, and approved the final draft.
- Wei Sun, Hui-Hua Wan and Junwen Zhai conceived and designed the experiments, authored or reviewed drafts of the paper, and approved the final draft.

### Data Availability

The data are available at NCBI: SAMN16927499, SAMN16927500, SAMN16927501, SAMN16927502, SAMN16927503, SAMN16927504; PRJNA680948.

Additional raw measurements are available in the Supplemental Files.
The unigene data are available at NCBI: GJQF00000000.

## Supplemental Information

Supplemental information for this article can be found online at http://dx.doi.org/10.7717/peerj.13106#supplemental-information.

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
