# Peer review of "Integration of the metabolome and transcriptome reveals indigo biosynthesis in Phaius flavus flowers under freezing treatment"

_PeerJ, doi:10.7717/peerj.13106_

## Round 0.1 · original submission · Major Revisions

Dear Drs. Wan and Zhai,

As you can see, all the reviewers found that your work was interesting.
However, they provided several comments and suggestions to strengthen your manuscript.

Reviewer 1 provided lots of valuable comments. For example, one of
his/her concerns is about the description of the Materials & Methods.
Reviewer 2 had several important questions, for example freezing
treatment and your experimental design, lacking information of the
condition of cells, and the purpose of this experiment.

I would like to ask you to address or to respond with reasons not to
follow the suggestion made by these reviewers.


Best regards,
Atsushi Fukushima

Reviewer 1 ·

Basic reporting

Needs significant correction in text at various places, grammars and updating the literature.

Experimental design

- Manuscript has tried to make some advancement in indigo biosynthesis in plants specifically in Phaius flavus a monocot.
- Materials & Methods: Plant material - This section is rather more ambiguous. Needs justification for experimental setup and starting material for downstream analysis.

Validity of the findings

- Data values needs to be re-check and verified including in supplementary tables.
- Speculation about the experimental treatment in conclusion section needs to be revised.

Additional comments

 Manuscript entitled “Integration of the metabolome and transcriptome reveals indigo biosynthesis in Phaius flavus flowers under freezing treatment (#55602)” has tried to make some advancement in indigo biosynthesis in plants specifically in Phaius flavus, a monocot.

 Overall manuscript and data presentation is satisfactory and needs correction in text at various places, grammars and updating the literature, points needs attention are listed below (also find in attached PDF file);

1 Introduction; line 62 “plants causes blue-changing in short time.”
2 Line 72 “ protection of its”
3 Line 75 “(Daykin, 2011) When the” needs full stop
4 Line 81 “indigo, indigo” delete repeats
5 Line 87-90 “Re-write the sentence as follows - The transcriptomics approach has been used to analyze the global expression of genes related to indigo biosynthesis. Further, we used the real-time quantitative polymerase chain reaction (qPCR) to verify the key gene expression patterns of indigo biosynthesis before and after the freezing treatment.”
6 Line 95-104 Materials & Methods: Plant material - This section is rather more ambiguous. Re-write this section to make it clear to justify the experimental setup and starting material for downstream analysis.
7 Line 98-99 “full blooming period - mention the time (date/ months)”
8 Line 100-102 experimental plan of freezing treatment is not realistically justified. Why not the flowers were sampled from plants during the lowest temperature condition OR kept inside the cold room like facility at 4-7 degree C (fridge compartments). Treatment with liquid nitrogen does not resemble the natural chilling/freezing conditions.
9 Line 104 “each plant part” - Flower is the only plant part was taken for analysis and presented in Fig. 1. What is the meaning of "each plant part"?
10 Line 104 “Add your materials and methods here” delete the sentence.
11 Line 105 correct metabolomics to metabolites
12 Line 105 correct metabolomics to metabolomic
13 Line 117 check the grammar “was used” instead of “are”
14 Line 119-120 what is “B” after percent unit
15 Line 133 correct “come” to “were procured/ obtained”
16 Line 138 “P. flavus powder” not the plant powder but the “P. flavus flower powder”
17 Line 139-140 “Dry 1 ml ----- methanol solution” Write this sentence in past tense.
18 Line 143 correct the “indicant”
19 Line 146-147 what is “B” after percent unit
20 Line 150 correct the “indicam”
21 Line 152-153 (Bai et al., 2019)[40]”, line 178-179 (Bryant et al., 2017)[41], line 200, 211 ---- etc., follow the journals guideline of citations.
22 Line 154-155 Correct “Before injecting the sample for GCMS-QQQ analysis, it was passed through a 0.45μm membrane filter.”
23 Line 157 “split injection split ratio” not clear
24 Line 158 at “the” rate of
25 Line 164 correct “Extraction of RNA was carried out from P. flavus flowers”
26 Line 168-170 What is the difference between RNA concentration 100 ng/ul and total amount 5ug. Re-write this section.
27 Line 195 “P. flavus flower,”
28 Line 199 “edgeR” Original author for edgeR package is "Robinson MD, McCarthy DJ, Smyth GK (2010). “edgeR: a Bioconductor package for differential expression analysis of digital gene expression data.” Bioinformatics, 26(1), 139-140. doi: 10.1093/bioinformatics/btp616"
29 Line 208-209 Re-write “The specificity of ------ total RNA”
30 Line 230 “flavonoids” remove the hyperlink
31 Line 244 “freezing treatment. (P < 0.05).” remove full stop
32 Line 247 “synthesis” change to “biosynthesis”
33 Line 260-261 “which 121,802 (53.12%) unigenes were annotated in the NR database;” Re-check the value against respective database and table 1.
34 Line 269-272 Recheck the values. Number of unigenes annotated against each category (Cellular processes - 65264; metabolic processes - 59455 and biological regulation - 23744), do not corroborate with total number of unigenes under GO functional annotation.
35 Line 286 (Fig. 4A) What is the difference between Fig. 4A and supplementary figure 1, in terms of sequencing analysis and ontological distribution?
36 Line 293-297 Number of unigenes mapped against KEGG database; do match with total number of gene count in Table S1.
37 Line 310-311 (Fig. 5 and supplementary table S3) Delete.
38 Line 312-315, 363-364 Arrange the citations as per journals guideline.
39 Line 320 & 321 what was the basis for selection of unigenes [CYP71A4 (Cluster-38184.135548), GLU (Cluster-38184.103035)] for qRT-PCR analysis ?
40 Line 367 “identify” change to “identified”
41 Line 375 Check for relevance of the supplementary table to the sentence.
42 Line 397 Supplementary Table S2 is listed the primers detail. Check for relevance of the supplementary table to the sentence.
43 Conclusion section: Line 422-423 “Therefore, freezing treatment ------ indigo-related compounds.” In general, physical (mechanical ) and biological (fermentation) treatment is given to break cells to release the cell content, feasibility of freezing treatment/injury to tons of biomass seems to be costly affair.
44 References Section: In most of the references, list of authors is incomplete. Last author separated by "and", missing in other references. Follow the journal guidelines.

Check all references for correctness; few examples are given below;

45 Line 429 Adachi, J., 2001. Line 440, 443, 472 Incomplete author list. Follow the journal guidelines.
46 Line 432 Hubei Agric Sci. 58(09), 107-109 ”+113”???.
47 Line 450-452 Re-check the reference for correctness.
48 Line 456-457 Volume number is missing
49 Line 461 and 467 Huang et al., References not verifiable.
50 Line 464 Correct either “and” or “&” and follow the same consistency throughout the manuscript and reference list.
51 Line 497-498 remove hyperlink from authors name
52 Line 524 Check the reference for correctness.

Summarizing, manuscript needs major correction prior to acceptance for publication.

Annotated reviews are not available for download in order to protect the identity of reviewers who chose to remain anonymous.

Reviewer 2 ·

Basic reporting

In Line 103 -104, “ Three samples were prepared for each plant part as replicates (Fig. 1)Add your materials and methods here.”
There is a mistake in the sentence.

In Line 140, “…was performed using the indigo, indirubin, indicant, and the indole quantitative reference…” indicant → indican

In Line 150, “….263.0815→219; indicam 296.1129→134.”
indicam → indican

In Line 394, “eventually forming indigo (Epstein et al., 1967; Minami et al., 1999). Studies”
Please enter "." after "...1999)"

In Line 421, “such as indole, indigo and indirubin in P. flavus were….” Please enter "," after "...indole".

In Line 389-390 & 390-391, “the gene expression level of indican…”
Indican is not gene.

Experimental design

1. How long perfom you the freezing treatment? Just flash by liquid nitrogen?
How treatment method with liquid nitrogen did the authors perform?
Immerse? Flash?
You should describe a more detailed method. It is crucial.

2. After the freezing treatment, the cells were alive? The cells were frozen?
If the cells died, it is a normal phenomenon to form indigo.
You should describe the condition of cells.

3. If the cells died… After the freezing treatment, is it possible that the cells can biosynthesize the metabolites? It seems to be impossible?
Please explain why you think the metabolites are biosynthesized.

Validity of the findings

You said that the increase of indole resulted from the promotion of metabolism by some enzymes such as TSA (in Line 366-370).
It is thought a possibility of degradation of indole derivatives such as indican?
Please explain why you think that indole increased by the metabolism.

In Line 382, “The reason is that indoxyl has a toxic effect on plants.”
Indoxyl is a toxic. It is true?
Indigo forming from indoxyl is precipitate, so indigo is any problem in the cells.

Additional comments

The data of transcriptome and metabolomics are well quantified, organized, and described. However, the condition of cells after freezing was not explained. At present, I cannot tell whether the freezing treatment is reasonable or not. Please describe more detail of the method.

I think that you need to describe the purpose of this experiment in more detail.

---

## Round 0.2 · Minor Revisions

Dear Drs. Wan and Zhai,

The revised manuscript has been evaluated by the original reviewers. As you will see, both reviewers pointed out minor essential issues. Please re-revise the manuscript accordingly.

Best regards,
Atsushi Fukushima

Reviewer 1 ·

Basic reporting

Most of the comments have been replied/ resolved satisfactorily. And accordingly the MS is improved.

Experimental design

Material and method parts needs revision which is suggested in comments. Most of the comments related to experimental design and sample treatments have been replied.

Validity of the findings

No comments

Additional comments

Manuscript entitled “Integration of the metabolome and transcriptome reveals indigo biosynthesis in Phaius flavus flowers under freezing treatment (#55602-v1)”

 MS now seems to be improved with some clarity, however it still needs minor corrections as listed below (also find in attached PDF file);

1 Materials & Methods: Plant material - Line 103-104 Re-write as follows “The flowers without treatment were used as the control group (CK).”
2 Line 105 replace “RNA sequencing” with “RNA isolation”
3 Line 168-176 rewrite paragraph as follows “Extraction of RNA were carried out from treated and control P. flavus flowers using the plant RNA rapid extraction kit (Beijing Adler Biotechnology Co., Ltd., Beijing, China; RN38 EASYspin PLUS) as per the manufacturer’s instructions. The concentration and purity of the RNA were measured with the Nanodrop 2000 spectrophotometer, and the Agilent 2100 instrument was used to assess RNA integrity. Samples with RNA integrity number (RIN) > 7 and concentrations ≥ 100 ng/μL were taken for cDNA library construction. cDNA library construction was performed using 5µg total RNA and NEB Next Ultra RNA Library Prep Kit from Illumina. The cDNA library was checked for quantity and quality with Qubit 2.0 (Life Technologies, USA) and Agilent Bioanalyzer 2100 system respectively. Then sequencing was carried out using Illumina HiSeq 2500 platform.”
4 Line 214-215 Re-write as follows “Gene-specific primer pairs for 150-250 bp regions were designed for different genes (Supplementary Table S1).”

Check all references for consistency; few examples are given below;

1 Line 436-438, Line 470,... etc. Check for space in name initials. Check throughout the reference list.
2 Line 461 “Garcia-Macias, P., and John, P., 2004.” use of "," and along with full stop, "and" delete
3 Line 471-472 Abbreviate the journal name to "J. Mol. Catal. B Enzym."
4 Line 478 correct the authors name “Thul S T, Sarangi B K,”
5 Line 493 and 495 Check the style of mentioning authors (See PDF).
6 Line 505 “Plant Cell Physiol”
7 Line 521, 532, 556 write journal names in italics.

Annotated reviews are not available for download in order to protect the identity of reviewers who chose to remain anonymous.

Reviewer 2 ·

Basic reporting

The sentences added by the authors are reasonable and include the needed information.

Experimental design

The same comment with “Basic reporting”.

Validity of the findings

The same comment with “Basic reporting”.

Additional comments

I have understood your answers to my question.
However, I think that the experimental design of frostbite may be hard to understand.
I recommend that you should add some more explanation in the discussion or other sections. For example, as you said in your answer, “after being frozen for 2 seconds, the P. flavus flowers cells damaged due to frost injury which may disrupt the lipid…………, but the DNA, RNA, and protein were all alive, which could express……”.

---

## Round 0.3 · Minor Revisions

Dear authors,

Thank you for revising. However, Reviewer 2 still had some minor problems. Would you please carefully revise the manuscript to address the comments raised?

Best regards

Reviewer 2 ·

Basic reporting

The sentences added by the authors are reasonable and include the needed information.

Experimental design

The same comment with “Basic reporting”.

Validity of the findings

The same comment with “Basic reporting”.

Additional comments

This article is well revised, but I recommend a little more modification for a better one.

1. The content of line 101~103 may not fit in “Material & Methods”.
Therefore, the sentence should be added in the last paragraph of “Introduction” or the “Discussion”.

2. Line 73~74 in “Introduction”
The authors described the enzyme to form indoxyl as CYP450. Recently, Inoue et al. reported the possibility that Flavin-monooxygenase may catalyze the hydroxylation of indole. Therefore, I recommend touching on the possibility in the sentence. (An indigo-producing plant, Polygonum tinctorium, possesses a flavin-containing monooxygenase capable of oxidizing indole. Biochem. Biophys. Res. Commun. 534, 199-205 (2021))

---

## Round 0.4 · Minor Revisions

Dear authors,

Thank you for revising. However, you should carefully revise the manuscript according to the following comments.

> the Section Editor has commented and said:
>
> "The manuscript concentrates on only a small collection of candidate genes related to indigo biosynthesis; however, with a concrete set of conditions used the full collection of Trinity assembled constructs should be provide at a given resource for reference; the raw sequencing data does not promise identical assemblies if done by another group. The assembly should be made available at a third-party resource as untapped DEGs may have been missed; and reported annotations are lost as these are not presented. We are only provided annotations for the very small set of candidate surveyed negating much of the research effort provided. The manuscript should be approved once the assembly set can be made available in some form; possibly with missing annotations mentioned in the manuscript. Basically we are only provided annotations for about 74 sequences. Below are other noted suggestions detected in the manuscript:
>
> EDITS
> LINE NO: / BEFORE / AFTER / [COMMENTS]
> LINE 35: / we further candidate / we further screened candidates for / [.]
> LINE 59: / . / . / [.]
> LINE 66: / (Polygalaceae). Base group / (Polygalaceae), and a base group / [.]
> LINE 70: / (Orchidaceae), etc / (Orchidaceae), among others / [.]
> LINE 76: / indican and stored / indican and is stored / [.]
> LINE 109: / treatment set 3 / treatment set of three / [.]
> LINE 161: / was rise to / was raised to / [.]
> LINE 162: / then rise to / then raised to / [.]
> LINE 167: / P. flavus / <i>P. Flavus</i> / [ italicize here.]
> LINE 201: / folower, / flowers, / [.]
> LINE 236: / derivates, Phenolamides / derivatives, Phenolamides / [.]
> LINE 236: / derivatives, Others, of / derivatives, and other of / [.]
> LINE 269: / Termbl / Trembl / [.]
> LINE 288: / total of 171,34 / total of 17,134 / [.]
> LINE 334: / screened out before / screened before / [.]
> LINE 338: / derivatives, Others. / derivatives, among others. / [.]
> LINE 339: / compound in plant immune / compound in the plant immune / [.]
> LINE 347: / MetWare database / MetWare database (CITATION) / [needs citation.]
> LINE 384: / Inoue et al. have / Inoue et al. (2021) / [.]
> LINE 385: / hydroxylation (Inoue et al., 2021). / hydroxylation. / [see line 384.]
> LINE 401: / reversed. / reversed, COMPLETE SENTENCE. / [incomplete statement.]
> LINE 410: / genes are concerning to indole / genes related to indole / [.]
> LINE 428: / study screened out several / study screened several / [.]
> LINE 429: / a candidate basis / a pool of candidates / [.]"

Best regards

---

## Round 0.5 · Minor Revisions

Dear authors,

Thank you for your reply. Would you please revise the manuscript according to the following comments by our Section Editor?

"Google drive is not necessarily what I would call third-party as the authors can revoke permissions. Something like NCBI, figshare, or others would be more acceptable as these are managed data archives. This was the most obvious issue; however, I did request more valuable connections between annotations and sequences. I hope the authors pay more attention to detail in their revisions."

Best regards

---

## Round 0.6 · Minor Revisions

Dear Dr. Zhai and co-authors,


Thank you for your revision. However, our Section Editor has commented as follows. Would you please revise the manuscript?

Best regards

--
"The data provided at NCBI as SAMN22590284 is simply the raw data, not the assembled data. The authors should not expect the readers to download the raw data and assemble them again, as a second assembly would not match the data as presented given subtle differences in compute environments.

Assembled transcriptomes can be deposited as a transcriptome shotgun assembly (GenBank TSA resource). I am displeased that the authors do not seem familiar with the data they are reporting on. There is still the need to provide assembled sequence data which matches that described in the manuscript.

Revision is still required. If I am wrong please correct me. The Trinity data of the P. flavus SRX12789523 is raw data of Illumina HiSeq 2000 data, not the final assembly!"

---

## Round 0.7 · accepted · Accept

Dear authors,

Thank you for your revision.

Best regards